# Trends in Phenolic Profiles of *Achillea millefolium* from Different Geographical Gradients

**DOI:** 10.3390/plants12040746

**Published:** 2023-02-07

**Authors:** Jolita Radušienė, Birutė Karpavičienė, Lina Raudone, Gabriele Vilkickyte, Cüneyt Çırak, Fatih Seyis, Fatih Yayla, Mindaugas Marksa, Laura Rimkienė, Liudas Ivanauskas

**Affiliations:** 1Laboratory of Economic Botany, Nature Research Centre, Akademijos Str. 2, 08412 Vilnius, Lithuania; 2Department of Pharmacognosy, Lithuanian University of Health Sciences, Sukileliu Av. 13, 50162 Kaunas, Lithuania; 3Laboratory of Biopharmaceutical Research, Institute of Pharmaceutical Technologies, Lithuanian University of Health Sciences, Sukileliu Av. 13, 50162 Kaunas, Lithuania; 4Vocational High School of Bafra, Ondokuz Mayis University, Samsun 55200, Turkey; 5Department of Field Crops, Faculty of Agriculture and Natural Sciences, Recep Tayyip Erdoğan University, Rize 53100, Turkey; 6Department of Biology, Faculty of Arts and Sciences, Gaziantep University, Gaziantep 27310, Turkey; 7Department of Analytical and Toxicological Chemistry, Lithuanian University of Health Sciences, Sukileliu Av. 13, 50162 Kaunas, Lithuania

**Keywords:** *Achillea millefolium*, caffeoylquinic acids, flavonoids, intraspecific variation, latitude, wild populations

## Abstract

The traditional widely used raw material of *Achillea millefolium* is currently mainly derived from wild populations, leading to diversification and uncertainty in its quality. The aim of the study was to determine the accumulation differences of phenolic compounds between geographically distant populations of *Achillea millefolium* from northern and southern gradients. Plant material was collected from Gaziantep and Nevşehir provinces in Turkey and from wild populations in Lithuania. A complex of nine hydroxycinnamic acids and eleven flavonoids was identified and quantified in the methanolic extracts of inflorescences, leaves, and stems using the HPLC-PDA method. Caffeoylquinic acids predominated in leaves, while inflorescences tended to prevail in flavonoids. The PCA score plot model represented the quantitative distribution pattern of phenolic compounds along a geographical gradient of populations. The content of phenolic compounds in plant materials from northern latitudes was more than twice that of plants from southern latitudes. A significant correlation of individual phenolic compounds with latitude/longitude corresponded to their differences between two countries. Differences in accumulation of caffeoylquinic acids and flavonoids revealed several intraspecific groups within *A. millefolium*. Our findings suggest that spatial geographical data on the distribution of phenolic compounds in *A. millefolium* populations could be used as a tool to find potential collection sites for high-quality raw materials.

## 1. Introduction

*Achillea millefolium* L. s.l., commonly known as yarrow or milfoil, is a heterogeneous group of closely related taxa of perennial herbs widely distributed throughout the temperate and boreal zones of the Northern Hemisphere [1,2]. *Achillea millefolium* is common in central and northern Europe and sparsely found in southern Europe [3]. In this regard, Turkey is the largest area of *Achillea* species, with 69 species identified that have adapted to a wide variety of habitats from desert and mountains to coastal regions [4]. In the identification of individual yarrow species, it is important for both collectors and processors to be aware of the potential quality of the raw material. Moreover, chemical profiling of specialized metabolites in plants is the basis for further chemophenetic studies. Yarrow is one of the oldest medicinal plants and has been used in folk medicine for thousands of years; its pollen was found in a 65,000-year-old of *Homo neanderthalensis* tomb in the Shanidar cave in Iraq [5]. Yarrow raw material (*Millefolii herba*) is included in the European Pharmacopoeia and described by the European Medicines Agency as a traditional drug [6]. The therapeutic indications of *A. millefolium* have been well documented in numerous clinical studies. *Achillea millefolium* extracts have strong anti-inflammatory, anti-arthritic [7], antifungal [8], antispasmodic [9], antibacterial [10], hypotensive, cardiovascular, bronchodilator [11], and analgesic [12] effects. A wide variety of specialized metabolites of *A. millefolium* are considered to be responsible for the curative properties and therapeutic applications of plant extracts [13,14]. Pharmacological efficiency in yarrow is mostly attributed to phenolic compounds and essential oils, the most important of which are caffeoylquinic acids, flavonoids, and sesquiterpene lactones, which contribute to the main multifunctional biological activity [15,16,17]. Different polyphenolics have been identified, including hydroxycinnamic and hydroxybenzoic acids [18,19,20,21,22,23], along with different flavonoids [19,21,22,23,24].

*Achillea millefolium* raw material is mainly derived from wild populations, which leads to the diversity and uncertainty of raw material specification requirements. The variety of plant material quality is considered to be a consequence of environmental effects, reflecting the synthesis of specialized metabolites and plant adaptation to local conditions [25]. However, it is difficult to identify one or a few environmental factors that regulate the synthesis of chemical compounds in plants. In this regard, geographical gradients represent a broad range of environments, especially along latitudes from north to south. Over altitude and latitudinal–longitudinal gradients, multiple environmental factors, including total solar and UV-B radiation, temperature, precipitation, nitrogen availability, aridity indices, soil fertility and acidity, exposure, and even land-use history, varied substantially [26]. Consequently, geographical gradients can be treated as factors generating environmental effects and are expected to influence specialized metabolism in plants. The intraspecific diversity of phenolic compounds accumulation in individual species along their geographical gradients is poorly known. In this context, we hypothesized that geographical gradients have a significant influence on the accumulation of phenolic compounds in wild populations of *A. millefolium* and can be used as a tool to identify high-quality raw material harvesting sites. The study aimed to establish the accumulation trends of phenolic compounds between geographically distant populations of *A. millefolium* in Turkey and Lithuania and to evaluate their differences in plant organs. To the best of our knowledge, there are no available data on differences in the accumulation of phenolic compounds in plant organs of *A. millefolium* depending on the geographical gradient of the population site. The presented study has the potential for planning the targeted use of plant materials and managing the overexploitation of plant resources.

## 2. Results

### 2.1. Phenolic Profiles of Inflorescences, Leaves, and Stems in Turkish and Lithuanian Populations

A total of nine phenolic acids and eleven flavonoids complexes were identified and quantified in terms of the phenolic profiles of inflorescences, leaves, and stems of *A. millefolium*. Significant differences in the accumulation of individual phenolic compounds were found in leaves, inflorescences, and stems from Turkish and Lithuanian populations (Table 1). Chlorogenic acid and 3,5-*O*-dicaffeoylquinic acid were predominant in all plant organs and populations, with the highest content in leaves, followed by inflorescences and stems. 1,5-*O*-dicaffeoylquinic acid was one of the main phenolics in leaves, inflorescences, and stems of Lithuanian *A. millefolium*. Meanwhile, in Turkish populations, 1,5-*O*-dicaffeoylquinic was detected in small amounts only in leaves and stems, while inflorescences did not accumulate this compound. Another phenolic compound with noteworthy amounts in both leaves and inflorescences from both locations was 3,4-*O*-dicaffeoylquinic acid. Small amounts of caffeic acid were detected in all plant organs in populations from both countries, while 1,3-*O*-dicaffeoylquinic acid was found only in leaves and stems of Lithuanian populations. Meanwhile, the content of 4-*O*-caffeoylquinic acid did not differ between plant organs for all populations, but the leaves of Turkish populations accumulated significantly higher amounts of this compound than in Lithuania.

Meanwhile, inflorescences of *A. millefolium* accumulated the highest amount of flavonoids, followed by leaves and stems. Among the flavonoids, flavones luteolin, apigenin, and their derivatives dominated in inflorescences. Luteolin-3,7-*O*-diglucoside was detected only in inflorescences and leaves of Turkish populations and has not been found at all in Lithuanian populations. The identified flavonol complex in *A. millefolium* was represented by quercetin and its derivatives, namely quercitrin, rutin, isoquercitrin, and methylated flavonol santin. A high level of rutin was detected in leaves, followed by stems and inflorescences. Santin was found in all plant organs, with the highest content in inflorescences. Meanwhile, quercetin and quercitrin were presented in minor amounts and did not expose significant differences between plant organs. The small amount of isoquercitrin differed significantly between plant organs and populations, with the highest amount found in inflorescences of Turkish populations, but not detected in their stems. Small amounts of phenolics were found in the stems of all populations, with more notable levels of chlorogenic acid and rutin in populations from both countries. Overall, the accumulation pattern of caffeoylquinic acids and flavonoids in *A. millefolium* populations showed significant quantitative differences between plant organs for most individual compounds, except for 4-*O*-caffeoylquinic acid, quercitrin, and quercetin, whose amounts did not differ significantly.

Comparison of the phenolic profiles of *A. millefolium* populations between the two countries revealed qualitative differences in the accumulation of 1,3-*O*-dicaffeoylquinic acid, quercitrin, and luteolin 3,7-*O*-diglucoside. 1,3-*O*-dicaffeoylquinic acid and quercitrin were detected only in populations from Lithuania, while luteolin 3,7-*O*-diglucoside was found in plants from Turkey. All other compounds were found in *A. millefolium* populations from both countries with significant quantitative differences in all plant organs, except for rutin, the amount of which did not differ significantly between countries for all plant organs (Table 1). Furthermore, amounts of individual phenolic acids highly positively correlated with latitude/longitude, with the exception of 1,3-*O*-caffeoylquinic and 4-*O*-caffeoylquinic acids. The correlations between individual flavonoids and latitude/longitude were variable. A positive correlation of apigenin-7-*O*-glucoside with latitude/longitude was found for all plant organs, while the amounts of isoquercitrin, luteolin-*O*-3,7-diglucoside, and santin in inflorescences and quercetin in leaves and stems showed a negative correlation with geographical gradients (Appendix A).

Furthermore, the Turkish populations originated from Southeastern and Central Anatolia geographical regions. The sites of populations in the Central Region were at a higher latitude and elevation than sites from Southeast Anatolia Region. Significant differences between the two Anatolian regions were found in the levels of 3,5-*O*-dicaffeoylquinic acid in inflorescences and stems, luteolin and luteolin-7-*O*-glucoside in inflorescences, santin in leaves, and 3,4-*O*- and 4,5-*O*-dicaffeoylquinic acids and luteolin-7-*O*-glucoside in stems. On the other hand, no significant correlation was found between the total phenolics content and site elevation (Appendix A). Nevertheless, the observed differences in compounds’ accumulation between the two regions did not reveal a reasonable dependence along latitude/longitude gradients. In general, the significant correlation of individual phenolic compounds with latitude and longitude corresponded to their significant differences between two countries.

### 2.2. Total Phenolic Acids and Flavonoids from Two Locations

Differences in the accumulation of phenolic compounds in plant organs between the two geographical areas were more evident when all compounds were grouped into caffeoylquinic acids and flavonoids and presented as total relative values to total phenolics. In this way, leaves were significantly dominated by caffeoylquinic acids, accounting for an average of 79.9 and 88.2% of the total phenolics in the Turkish and Lithuanian populations, respectively, followed by stems (69.0 and 84.4%, respectively) (Table 2). Meanwhile, the values of phenolic acid and flavonoids did not differ significantly in inflorescences of Turkish populations. On the other hand, phenolic acids significantly prevailed in the inflorescences of Lithuanian populations, compared with flavonoids (Table 3). Moreover, the mean ratio of total flavonoids in all plant organs was significantly higher in southern populations than in northern populations, while phenolc acids increased significantly at higher latitudes. Mean values of total phenolic acids differed significantly between countries, while flavonoids showed differences only in leaves. Overall, the Lithuanian populations accumulated a higher total content of identified phenolic compounds in all plant organs, compared with the Turkish populations.

### 2.3. Chemical Variation of Populations: Principal Component Analysis (PCA)

A two-dimensional PCA1 score plot model explaining 59.50% of the total variance of the dataset was applied to visualize the similarities and differences among the studied populations using selected phenolics as statistically independent variables (Figure 1).

The PCA1 results based on eigenvalues correlation matrix with PC1 and PC2 (21.84 and 2.54, respectively) are presented in Table 4. PC1 explained 53.3% of the total variance of the dataset and showed strong negative correlations with caffeoylquinic and caffeic acids in all plant organs and variable correlations with selected flavonoids (Table 4). PC2 accounted for only 6.2% of the total variance and was positively correlated with caffeic acid in leaves and negatively with luteolin-7-*O*-glucoside in inflorescences and santin in stems.

PCA1 score plot pattern showed the arrangement of all populations into two groups scattered on opposite sides of the square plots. The group located on the right-hand score plot along the positive PC1, close to the zero point, clustered most of the Turkish populations, which accumulated from the lowest to average amounts of compounds within their range of variations. Variables with high positive loadings on PC1 that had the highest impact on the grouping of the Turkish populations were luteolin-*O*-3,7-diglucoside, isoquercitrin, and santin in inflorescences and 4-*O*-caffeoylquinic acid and quercetin in leaves and quercetin in stems. The variables showed a weak correlation with PC2 and, consequently, a weak contribution to the arrangement of populations, except for a high correlation with caffeic acid in leaves and santin in stems. The distant location of populations No. 1 and 10 indicated that the amounts of phenolics in their plant materials differed highly from other Turkish populations. The remote position of population No. 1 can be explained by the lowest amount of caffeoylquinic acids in all plant organs and the highest amount of santin and luteolin in inflorescences. The distant position of population No. 10 was due to the highest amount of luteolin and apigenin glucosides and luteolin-*O*-7-rutinoside in inflorescences and leaves. However, the PCA model showed no differences in the distribution of populations depending on the elevation of the site, which ranged from 639 to 1287 m.a.s.l.

The group on the left-hand score plot brought all *A. millefolium* populations from Lithuania, with exception of No. 18. In contrast to the first group previously discussed, the variables with high negative loadings on PC1 had the greatest impact on the distribution of these populations into several clusters. The cluster along the negative PC1 near the zero point combined the populations that accumulated from the lowest to the average amounts of compounds in their range of variation of Lithuanian populations. The arrangement of the second group is associated with high negative loadings of chlorogenic and 3,5-*O*-dicaffeoylquinic acids and quercitrin in inflorescences and leaves, and of luteolin-7-*O*-rutinoside in leaves, the amounts of which were the highest in the raw materials of the corresponding populations. The clustering of three populations (No. 17, 23, and 34) coincided with high levels of dicaffeoylquinic acids and luteolin- and apigenin 7-*O*-glucosides in their inflorescences and leaves. The remote position of population No. 25 is associated with high negative loadings of 4,5-*O*-dicaffeoylquinic acid in stems and positive caffeic acid in leaves. The position of population No. 18 on the right-hand score plot, next to the cluster of Turkish populations, showed their similarity, which corresponded to the lowest amounts of phenolic compounds in this population compared with other Lithuanian populations.

The PCA2 score plot model explaining 81.83% of the total variance of the dataset was applied to visualize the distribution of populations using the total content of phenolic acids and flavonoids as independent variables (Figure 2).

PC1 explained 65.8% of the total variance of the dataset and showed significant negative correlation with the total content of phenolic acids in all plant parts and significant negative correlations with the total content of flavonoids in leaves and stems. PC2 accounted 16.1% of the total variance and was negatively correlated with the total content of flavonoids in inflorescences (Table 5).

The PCA2 model showed the same tend as in PCA1, with the distribution of populations into distinct groups according to the country of their origin. On the other hand, the scores clustered less tightly on the PC plot, displaying greater variation than in the PCA1 model. The overlap position of some populations indicated their similarity. The overlap position of No. 18 with Turkish populations can be explained by the lowest amount of total phenolic acids compared with other Lithuanian populations. The position of population No. 26 on the right-hand score plot, next to the cluster of Turkish populations, is associated with the similarity of their total flavonoid content.

Consequently, PCA was used to summarize the results of intraspecific diversity of the studied *A. millefolium* populations from different geographical areas. The PCA score plot models represented the distribution of populations along geographical gradients. Intraspecific variability of *A. millefolium* based on the accumulation of phenolic compounds is considerably related to the environment of the growing site of the population.

## 3. Discussion

Chemical diversity of specialized metabolites is a rather common occurrence in vascular plants, induced by different exogenous and endogenous environmental factors. Over time, species adapt to the local environment through micro-evolutionary differentiation of populations, leading to the formation of new metabolome patterns expressed as intraspecific diversity [27,28]. Studies have shown that geographical gradients affect the synthesis of specialized metabolism, which determine the active adaptation of plant species to the environment at higher altitudes and latitudes [29,30]. The main factor associated with differences in phenolic compounds at high altitudes and high latitudes is the response to low temperatures or increased UV-B radiation, which generally induces photo-oxidative stress [31,32] and promotes the synthesis of phenolic compounds [33]. However, previous studies on different plant species have shown the variable effects of environmental gradients on phenolic metabolic pathways [34,35]. The opposite trend of accumulation of phenolic compounds, depending on the elevation, was found in *Vaccinium myrtillus* L. both for groups of compounds and for part of plants [36,37]. Nataraj et al. [38] reported that the concentration of phenolic compounds decreased in the leaves of *Artemisia brevifolia* Wall ex DC. with increased elevation. In our study, elevation was an insignificant environmental element at northern latitudes and was not applicable as a comparative factor between the two geographical areas. At the same time, phenolic compound accumulation and the elevation differences between the two Anatolian regions did not show a clearer trend to explain the differences in the accumulation of phenolic compounds between populations.

Large-scale latitudinal gradient studies have been explored on a limited number of different species, yielding controversial results. A positive correlation between flavone content and latitude was estimated for *Rutelia* species across large-scale latitudinal gradients [39]. Latitudinal and regional variations of flavonoid concentrations from south to north in Finnish *Betula pubescens* Ehrh. populations showed compound-specific latitudinal gradients associated with an increase in photoprotective quercetin and a decrease in apigenin derivatives towards low temperature areas in the north [40]. A strong positive effect of latitude was found on soluble phenolic and flavonol levels in *Vaccinium myritillus* leaves [36]. Meanwhile, a negative correlation between latitude and flavonoid accumulation was found in *V. myritillus* fruits [41] and *Juniperus communis* L. leaves [29]. It should be noted that the above studies were conducted in Finland, where northern latitudes and higher altitudes corresponded to subarctic growing conditions where plants were exposed to low average temperatures and high irradiance. Thus, our study contributes to the understanding of the increased accumulation of phenolic compounds at higher latitudes, excluding the effect of altitude as an insignificant factor in the studied northern areas. Furthermore, we observed opposite trends in the accumulation of caffeoylquinic acids and flavonoids along geographical gradients, when the relative value of phenolic acids increased in northern latitudes and flavonoids decreased compared with southern latitudes, and vice versa. Consequently, it can be assumed that the low temperature in northern latitudes has a greater importance for the accumulation of phenolic acids than for flavonoids. On the other hand, the trend of increasing total flavonoid content in southern latitudes can be explain by increased UV-B radiation at higher elevations compared with northern, suggesting a greater contribution of flavonoids to radical scavenging potential. In this regard, our findings are partially explained by previous studies showing the promotion of flavonoid synthesis at a high altitude due to increased UV-B radiation and oxidative stress [37].

According to Moore et al. [42], chemical changes should be sought where plants occupy new habitats and where climate and other factors have altered the biotic and abiotic environment. On the other hand, the response of organisms to environmental differences is often under genetic control [31]. Based on the presented data, it can be assumed that the intraspecific chemical diversity of *A. millefolium* populations from the same geographical location potentially reflected genetic differences in the exposure of the surrounding environment. Molecular studies of genotypes are the most appropriate reliable tool to explain the intraspecific chemical variability and available taxonomic differences, but the use of phytochemical studies can provide insight into further application of populations [43].

Complexes of caffeoylquinic acids and luteolin derivatives are the key components in the fingerprinting of all *A. millefolium* samples tested. Caffeoylquinic acids have recently attracted much attention owing to their numerous technological, biological, and pharmacological implications [17]. Furthermore, studies have supported the antibacterial and antiviral effects of caffeoylquinic acids [17,44]. In the context of the COVID-19 pandemic, luteolin and its derivatives, owing to anti-neuroinflammatory mechanisms, may be used to manage the post-COVID-virus condition. In its aglycone form, the flavone luteolin can penetrate into the brain and exert an anti-inflammatory effect on the target cells [45,46]. *Achillea millefolium* raw materials, with their specific profile, gain perspective as a matrix with hepatoprotective, neuroprotective, and cardioprotective activities mainly expressed by anti-inflammatory and antioxidant mechanisms [17,46,47]. Ali et al. [48] elucidated the mechanisms of the above activities with luteolin and chlorogenic acid as key contributors. Reproductive biological activity should be investigated by combining phytochemical and bioactivity markers and determining the phytogeographical and genotypic profiles of plant matrices.

The distribution of phenolic compounds in *A. millefolium* populations reported from various countries such as Iran [21], Turkey [20,49], Italy [18], Bulgaria [24], Central Europe [50], and Poland [51,52] is difficult to explain by geographical differences in locations. This may be because previous studies were conducted in limited areas and populations or without reference to plant organs, making them difficult to compare. However, in agreement with our results, it was reported that chlorogenic acid, 3,4-*O*-dicaffeoylquinic, and 3,5-*O*-dicaffeoylquinic acids, together with the flavonoid aglycones luteolin and apigenin and their glycosides, were the most common and major phenolics in *A. millefolium*.

According to Zidorn [53], correct taxonomic identification as well as information of geographical location, plant harvesting time, and plant organs, together with other indirect factors, were considered to be crucial in phytochemical studies. On the other hand, the extraction and method of chemical analysis may be factors leading to differences in the chemical composition of plant materials. Consequently, chemical studies of plant materials from different geographical regions using the same methods and the same equipment have many advantages as they reveal a more objective overall picture in the distribution trend of specialized metabolites. Moreover, chemical profiling of specialized metabolites in plants is the basis for further chemophenetic studies.

## 4. Materials and Methods

### 4.1. Plant Material, Sampling, and Identification

*Achillea millefolium* aerial plant material, from 16 different populations in Turkey and 20 from Lithuania, was collected during full flowering in June and August of 2019 and 2022. Plant material consisting of 30 single shoots per population was dissected into inflorescences, leaves, and stems and dried separately at room temperature. The botanical identification of species was performed on morphological characters according to descriptors [2,4]. The voucher specimens were deposited in the herbarium of Vocational High School of Bafra, Ondokuz Mayis University and Nature Research Centre, Institute of Botany.

### 4.2. Sampling Sites

Sampling was carried out in Gaziantep Province in Southeastern Anatolia Region (No. 1–9) and Nevşehir Province in Central Anatolia Region (No. 10–16) and in different districts in Lithuania (No. 17–36) (Figure 3). Data on plant sampling sites are presented in Table 6.

The Gaziantep Province is characterised by a transition between a Mediterranean and continental climate. The winter season is cold and rainy, while the summer is hot and dry. The long-term average temperature is 15.1 °C and the average annual precipitation is 463 mm, most of which falls in winter [54]. Sampling sites were covered between 37.2506–37.2667 °N and 37.1492–37.4663 °E, on calcareous mountain slopes and meadows at an elevation from 639 to 1011 m.a.s.l.

Nevşehir Province has a continental climate (semi-arid) with hot and dry summers and cold and rainy/snowy winters, with an annual rainfall of 415 mm, mainly in spring and winter. The long-term average temperature is 10.73 °C [55]. The sampling sites were located between 38.6167–38.7833 °N and 34 22 12.0–34.8663 °E, at an elevation of 954–1287 m.a.s.l., in open coniferous forest habitats.

Lithuania is situated on the edge of the East European Plain covering 56.27–53.53 °N and 20.56–26.50 °E. The country is in the temperate climate zone, in the subregion of Atlan tic-European continental mixed and broad-leaved forests. The long-term (1991–2020) average temperature is 7.4 °C. The average annual precipitation is 695 mm, most of which falls in summer. The growing season lasts 169 days in the east and 202 days in the west. The country is mostly flat, except for some hilly areas. The highest elevation is 297.84 m.a.s.l. Sampling sites were located between 54.09871–56.24353 °N and 23.72095–25.36114 °E, at elevations of 47–180 m.a.s.l., in dry or mesophytic grassland and pine forest habitats.

### 4.3. Chemicals

Analytical and HPLC grade solvents and reagents were used for chemical analyses. Acetonitrile (99.9%) and methanol (99.9%)as well as the references as quercetin, quercetin-3-*O*-glucoside, luteolin-7-*O*-glucoside, luteolin-7-*O*-rutinoside, rutin trihydrate, apigenin-7-*O*-glucoside, santin, 3-O-caffeoylquinic acid, 5-*O*-caffeoylquinic acid, 4-*O*-caffeoylquinic acid, 1,3-*O*-dicaffeoylquinic acid, 3,4-*O*-dicaffeoylquinic acid, 3,5-*O*-dicaffeoylquinic acid, 1,5-*O*-dicaffeoylquinic acid, and 4,5-*O*-dicaffeoylquinic acid, were obtained from Sigma-Aldrich (Steinheim, Germany); trifluoroacetic acid (≥99%) and apigenin were supplied from Fluka Chemika (Buchs, Switzerland). The purified deionized water (18.2 mΩ/cm) was produced using the Millipore Simpak1 Synergy 185 ultra-pure (Bedford, MA, USA) water system.

### 4.4. Extraction

The air-dried plant material was mechanically ground with a laboratory mill to obtain a homogenous powder. Samples of approximately 0.1 g (accuracy 0.0001 g) were extracted with 10 mL of 70% methanol at 40 °C for 30 min in an ultra-sonic bath. The extracts were filtered through 0.22 µm nylon syringe filters (Carl Roth GmbH & Co. KG, Karlsruhe, Germany) and stored at 4 °C until analysis.

### 4.5. Qualitative and Quantitative Analysis

The chemical analysis of plant material for the qualification and quantification of phenolic compounds was performed using a Waters Alliance 2695 mode system coupled with a 2996 PDA photodiode-array detector (Waters, Milford, MA, USA). The phenolic compounds were separated using ACE Super C18 (250 mm × 4.6 mm i.d., 3.0 µm) column (ACT, Aberdeen, UK), maintained at 35 °C. The mobile phase of binary gradient elution at a flow rate of 0.5 mL/min consisted of eluent A (0.1% trifluoroacetic acid in pure water) and eluent B (100% acetonitrile). The elution program was as follows: 0–40 min—10–30% B, 40–60 min—30–70% B, 60–64 min—70–90% B, and 64–70 min—90–10% B. The injection volume of the extract was 10 µL. Identification was performed in a range of 200–400 nm wavelengths by comparing UV/Vis spectral data and retention times to those of standard compounds. Validation data of the method with representative chromatograms are provided in our previous paper [56].

### 4.6. Data Analysis

Multivariate statistical analysis was performed using the software package Statistica 10.0 (StatSoft Inc., Uppsala, Sweden). One-way analysis of variance (ANOVA) was used to identify the significant differences of phenolic compounds among populations and plant organs. Significant differences were specified by a post-hoc Schefft’s test (*p* ≤ 0.05). Variable sets were compared pairwise by calculating the Pearson’s correlation coefficient (*r*) and Student’s *t* statistic. Differences in the ratio of groups of phenolic compounds between plant organs and between countries were determined using the Wilcoxon matched pairs test and the Kruskal–Wallis test. PCA1 was based on standardized variables that differed significantly between countries, including 15 inflorescence, 15 leaves, and 11 stem compounds. PCA2 was based on the sum of total phenolic acids and flavonoid standardized variables. Datasets of inflorescences, leaves, and stems were pooled together and used in PCA to obtain more compelling results for visualization.

## 5. Conclusions

Considering widespread application and commercial demand for yarrow raw material (*Millefolii herba*), which is currently mainly obtained from the wild, it is important for collectors and manufacturers to be aware of the quality and safety of the raw materials used in pharmaceuticals and dietary supplements. In this regard, screening of the intraspecific diversity of *A. millefolium* allows for the validation of product development from wild materials and helps to identify and select populations with a high potential for phenolic compounds.

This study indicated the importance of geographical gradients in the quality of phenolic compounds in plant organs of *A. millefolium*, emphasizing the importance of higher latitudes. The amount of phenolic compounds in plant materials from northern latitudes was more than twice as high as that in plants from southern latitudes, which may reflect greater plant adaptation to the respective conditions. Consequently, our study contributes to the understanding of the increased accumulation of phenolic compounds at higher latitudes. Opposite geographical gradient and accumulation trends of caffeoylquinic acids and flavonoids were found, where the relative value of phenolic acids increased in northern populations and the amounts of flavonoids decreased in southern populations, and vice versa.

Information on regional and geographical gradient range differences of phenolic compound profiles in *A. millefolium* will contribute to the knowledge of the distribution of specialized metabolites and their biochemical wild resources. The findings suggest that spatial geographical data on the distribution of phenolic compounds in *A. millefolium* populations could be used as a tool to find potential collection sites for high-quality raw materials.

The knowledge gained is important for the potential assessment of the environmental impact on the accumulation of phenolic compounds in medicinal plants, emphasizing the positive influence of northern latitudes. The obtained results have the potential for planning the targeted use of plant materials and for the conservation of plant resources from overexploitation.

## Figures and Tables

**Figure 1 plants-12-00746-f001:**
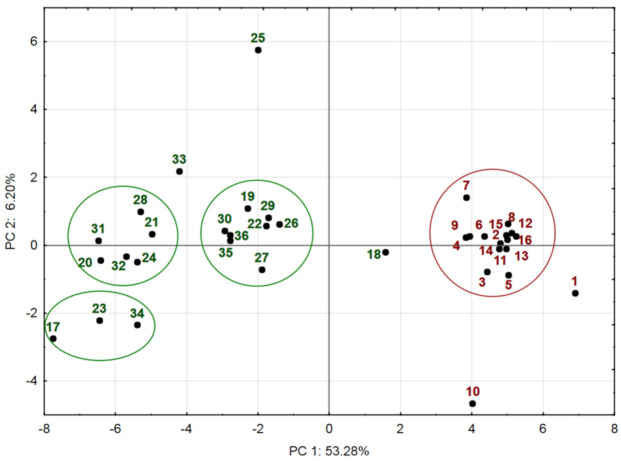
PCA1 score plot model representing the accumulation of phenolic compounds in *Achillea millefolium* plant materials from Turkey (No. 1–16) and Lithuania (No. 17–36) populations. Inflorescence, leaf, and stem datasets were pooled together and used for PCA. Scores marked and outlined in brown indicated Turkish populations; scores marked and outlined in green indicated Lithuanian populations.

**Figure 2 plants-12-00746-f002:**
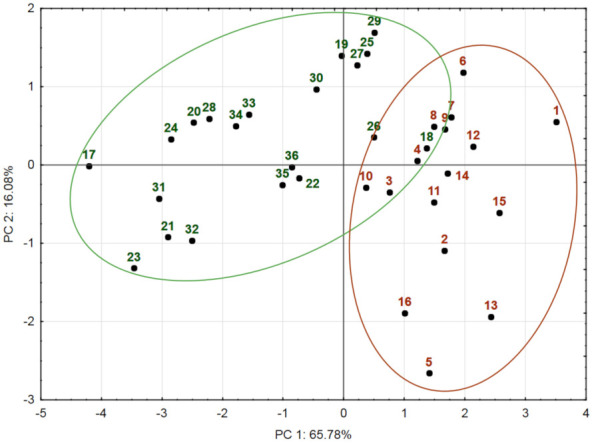
PCA2 score plot model representing the accumulation of total phenolic acids and flavonoids in *Achillea millefolium* plant materials from Turkish (No. 1–16) and Lithuanian (No. 17–36) populations. Inflorescence, leaf, and stem datasets were pooled together and used for PCA2. Scores marked in brown indicate Turkish populations; scores marked in green indicate Lithuanian populations.

**Figure 3 plants-12-00746-f003:**
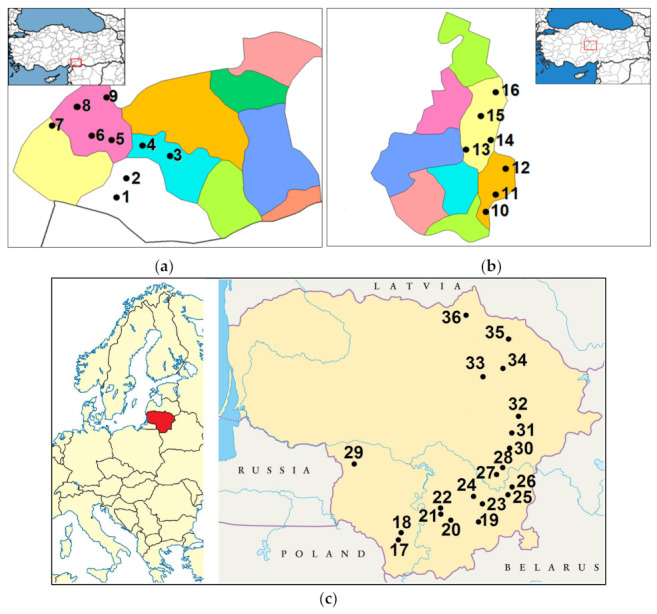
Maps of *Achillea millefolium* sampling sites in the provinces of Gaziantep (**a**) and Nevşehir (**b**) in Turkey and Lithuania (**c**).

**Table 1 plants-12-00746-t001:** The mean quantities (µg/g, DM) of phenolic compounds in inflorescences, leaves, and stems of *Achillea millefolium* in Turkey and Lithuania and comparison of their differences (*p* ≤ 0.05) according to *t*-test, as well as differences in compounds among plant organs for all populations by ANOVA (*p*_1_).

Compounds	Inflorescences (*n* = 36)					Leaves (*n* = 36)					Stems (*n* = 36)					F	*p* _1_
	Turkey		Lithuania		*p*	Turkey		Lithuania		*p*	Turkey		Lithuania		*p*		
	M	SD	M	SD		M	SD	M	SD		M	SD	M	SD			
Neochlorogenic acid	219.9	37.8	395.7	85.5	<0.001	557.3	282.4	979.8	261.5	<0.001	265.2	70.0	254.1	59.0	0.607	69.3	<0.001
Chlorogenic acid	1443.8	736.5	7735.3	2273.2	<0.001	4140.3	3294.7	19,483.3	6832.2	<0.001	1068.6	610.8	4069.3	1251.3	<0.001	27.6	<0.001
4-*O*-caffeoylquinic acid	1362.7	508.9	992.4	572.9	0.051	1536.7	686.6	408.3	478.9	<0.001	641.5	237.1	967.4	791.1	0.122	2.4	0.097
3.4-*O*-dicaffeoylquinic acid	1701.0	997.2	2582.3	690.5	0.004	982.9	869.4	3482.3	1231.2	<0.001	184.8	116.0	738.1	378.0	<0.001	30.7	<0.001
3.5-*O*-dicaffeoylquinic acid	2409.4	1078.6	8646.8	1750.2	<0.001	3749.8	2156.8	11,901.1	4087.7	<0.001	755.4	314.3	1448.4	465.3	<0.001	35.4	<0.001
1,3-*O*-dicaffeoylquinic acid	0	0	0	0		0	0	82.0	70.3		0	0	37.0	47.3		14.1	<0.001
1.5-*O*-dicaffeoylquinic acid	0	0	2345.6	954.3	<0.001	51.8	23.6	4924.7	2591.7	<0.001	64.4	25.5	1163.8	624.4	<0.001	10.2	<0.001
4.5-*O*-dicaffeoylquinic acid	318.4	125.5	1010.8	289.0	<0.001	142.8	93.5	1820.3	796.7	<0.001	105.0	62.5	505.2	284.7	<0.001	11.4	<0.001
Caffeic acid	1.6	6.4	19.6	13.9	<0.001	35.1	24.02	70.6	44.6	0.007	14.5	17.9	39.5	29.5	0.005	19.6	<0.001
Quercitrin	0	0	289.6	429.9	0.011	0	0	255.6	161.3	<0.001	0	0	90.0	56.6	<0.001	2.4	0.093
Rutin	188.5	120.4	168.7	180.4	0.709	1979.0	1340.7	2681.0	1259.8	0.116	649.4	369.2	872.6	511.2	0.152	69.5	<0.001
Quercetin	34.2	26.1	26.4	3.6	0.196	30.1	14.1	18.6	2.6	0.001	30.8	8.8	22.3	4.8	0.001	2.1	0.130
Isoquercitrin	425.0	432.2	6.5	20.7	<0.001	55.2	132.8	86.8	123.6	0.466	0	0	33.4	64.5	0.047	5.9	0.004
Luteolin	2699.7	2072.7	814.2	1126.9	0.001	395.2	915.7	146.2	57.2	0.232	120.3	36.5	134.4	9.8	0.107	20.3	<0.001
Luteolin-7-*O*-glucoside	1776.1	954.5	2772.1	702.5	0.001	262.3	485.7	975.5	670.0	0.001	48.7	51.6	94.2	98.2	0.103	106.4	<0.001
Luteolin-7-*O*-rutinoside	601.4	409.2	575.4	370.2	0.843	160.1	131.7	1154.4	511.8	<0.001	214.5	125.2	195.1	91.3	0.594	13.6	<0.001
Luteolin-3.7-*O*-diglucoside	1836.4	1507.8	0	0	<0.001	104.6	418.4	0	0.0	0.270	1.1	4.4	30.7	16.1	<0.001	11.9	<0.001
Apigenin	132.9	182.5	524.0	140.8	<0.001	19.2	55.2	9.0	11.6	0.425	0	0	0	0		65.1	<0.001
Apigenin-7-*O*-glucoside	241.9	215.8	3501.0	831.4	<0.001	23.0	91.9	214.3	103.1	<0.001	0	0	0	0		45.4	<0.001
Santin	502.4	231.3	222.8	57.31	<0.001	308.7	234.4	246.0	25.4	0.241	232.4	15.2	187.9	64.5	0.011	7.3	0.001
Total (sum of compounds)	15,895.4	4349.1	32,629.0	7223.8	<0.001	14,534.0	7047.0	48,939.8	14,818.8	<0.001	4396.5	1359.1	10,883.2	3080.0	<0.001	32.6	<0.001

**Table 2 plants-12-00746-t002:** Mean ratio (%) of total phenolic acids and flavonoids in plant organs of *Achillea millefolium* from Turkish and Lithuanian populations. Significant differences (*p*_1_) between compound groups according to the Wilcoxon matched pairs test and differences (*p*_2_) between countries according to Kruskal–Wallis test.

Plant Organ	Turkey		*p* _1_	Lithuania		*p* _1_	Acids	Flavonoids
	Acids	Flavonoids		Acids	Flavonoids		*p* _2_	*p* _2_
Flowers	48.4 ^a^	51.6 ^a^	0.501	73.8 ^a^	26.2 ^a^	<0.001	<0.001	<0.001
Leaves	79.9 ^b^	20.1 ^b^	0.001	88.2 ^b^	11.8 ^b^	<0.001	<0.001	<0.001
Stems	69.0 ^b^	31.0 ^b^	<0.001	84.4 ^b^	15.6 ^b^	<0.001	<0.001	<0.001

Values followed by the different letters differ significantly between plant organs, according to the Kruskal–Wallis test (*p* < 0.05).

**Table 3 plants-12-00746-t003:** Mean values (µg/g dry matter (DM)) of total phenolic acids and flavonoids in plant organs of *Achillea millefolium* from Turkish and Lithuanian populations. Significant differences (*p*_1_) between compound groups and differences (*p*_2_) between countries according to *t*-test.

Plant Organ	Turkey		*p*_1_ ^2^	Lithuania		*p* _1_	Acids	Flavonoids
	Acids	Flavonoids		Acids	Flavonoids		*p_2_*	*p_2_*
Flowers	7456.9 ^a 1^	8438.5 ^a^	0.371	23,728.4 ^a^	8900.6 ^a^	<0.001	<0.001	0.647
Leaves	11,196.7 ^b^	3337.3 ^b^	<0.001	43,070.4 ^b^	5869.4 ^b^	<0.001	<0.001	0.002
Stems	3099.3 ^c^	1297.1 ^c^	<0.001	9185.6 ^c^	1697.6 ^c^	<0.001	<0.001	0.056

^1^ Mean values followed by the different letters differ significantly between plant organs, according to the Scheffe’s test (*p* ≤ 0.05). ^2^ df—34.

**Table 4 plants-12-00746-t004:** PCA1 results based on Pearson correlation matrix between selected *Achillea millefolium* phenolic compounds in plant organs that differed significantly between countries and the two principal component scores.

No.	Variables	Inflorescences		Leaves		Stems	
		PC1	PC2	PC1	PC2	PC1	PC2
1	Neochlorogenic acid	−0.93	−0.12	−0.79	0.45	-	-
2	Chlorogenic acid	−0.96	−0.12	−0.92	0.21	−0.93	−0.08
3	4-*O*-caffeoylquinic acid	-	-	0.60	0.35	-	-
4	3,4-*O*-dicaffeoylquinic acid	−0.64	−0.39	−0.89	0.02	−0.89	−0.10
5	3,5-*O*-dicaffeoylquinic acid	−0.95	−0.13	−0.92	0.10	−0.88	−0.09
6	1,5-*O*-dicaffeoylquinic acid	−0.91	0.03	−0.85	−0.10	−0.86	−0.24
7	4,5-*O*-dicaffeoylquinic acid	−0.91	−0.02	−0.89	−0.04	−0.72	0.46
8	Caffeic acid	−0.67	0.16	−0.48	0.65	−0.59	0.45
9	Quercitrin	−0.55	−0.10	−0.86	−0.05	−0.78	−0.28
11	Quercetin	-	-	0.55	−0.34	0.52	−0.40
12	Isoquercitrin	0.58	−0.26	-	-	−0.62	−0.36
13	Luteolin	0.43	0.21	-	-	-	-
14	Luteolin-7-*O*-glucoside	−0.59	−0.59	−0.75	−0.44	-	-
15	Luteolin-7-*O*-rutinoside	-	-	−0.87	−0.07	-	-
16	Luteolin-3,7 *O*-diglucoside	0.65	−0.53	-	-	-	-
17	Apigenin	−0.78	0.30	-	-	-	-
18	Apigenin-7-*O*-glucoside	−0.95	0.00	−0.79	−0.49	−0.80	−0.20
19	Santin	0.64	−0.36	-	-	0.33	−0.77

**Table 5 plants-12-00746-t005:** PCA2 results based on Pearson correlation matrix between *Achillea millefolium* total phenolic acids and flavonoids in plant organs and the two principal component scores.

Variables	Inflorescences		Leaves		Stems	
	PC1	PC2	PC1	PC2	PC1	PC2
Phenolic acids	−0.92	0.16	−0.90	0.21	−0.93	0.19
Flavonoids	−0.36	−0.88	−0.85	0.02	−0.75	−0.28

**Table 6 plants-12-00746-t006:** Sampling sites, coordinates, elevation, and habitats of *Achillea millefolium* populations from Turkey and Lithuania.

No.	Province, Geographical Region	Latitude (°N)	Longitude (°E)	Elevation (m.a.s.l.)	Habitat
1	Nevşehir, Central Anatolia	38.6167	34.6162	954	Open coniferous woodland
2	Nevşehir, Central Anatolia	38.6500	34.6495	961	Open coniferous woodland
3	Nevşehir, Central Anatolia	38.6833	34.7162	1012	Open coniferous woodland
4	Nevşehir, Central Anatolia	38.7001	34.8663	1116	Open coniferous woodland
5	Nevşehir; Central Anatolia	38.7333	34.8829	1229	Open coniferous woodland
6	Nevşehir, Central Anatolia	38.7833	34.7329	1287	Open coniferous woodland
7	Nevşehir, Central Anatolia	38.7833	34.6162	1203	Open conifer woodland
8	Gaziantep, Southeast Anatolia	37.2506	37.1492	675	Roadside grassland
9	Gaziantep, Southeast Anatolia	37.2501	37.1828	778	Roadside grassland
10	Gaziantep, Southeast Anatolia	37.2334	37.1996	1011	Roadside grassland
11	Gaziantep, Southeast Anatolia	37.2333	37.2496	972	Calcareous mountain slope
12	Gaziantep, Southeast Anatolia	37.2334	37.3163	933	Calcareous mountain slope
13	Gaziantep, Southeast Anatolia	37.2500	37.3163	639	High altitude grassland
14	Gaziantep, Southeast Anatolia	37.2667	37.3663	937	High altitude grassland
15	Gaziantep, Southeast Anatolia	37.2667	37.4163	946	High altitude grassland
16	Gaziantep, Southeast Anatolia	37.2667	37.4663	861	High altitude grassland
17	Veisiejai, Lazdijai distr.	54.09871	23.72095	125	Dry grassland
18	Meteliai, Lazdijai distr.	54.28465	23.74717	114	Mixed forest, roadside
19	Pirčiupiai, Šalčininkai distr.	54.39087	24.95129	131	Pine forest, roadside
20	Einororys, Alytus distr.	54.44614	24.38971	120	Mesophytic grassland
21	Geruliai, Alytus distr.	54.53120	24.27056	130	Mesophytic grassland
22	Gojus, Prienai distr.	54.56845	24.28128	96	Dry grassland
23	Žuklijai, Trakai distr.	54.50986	24.68252	162	Mesophytic grassland
24	Pamiškė, Trakai distr.	54.62607	24.51166	180	Mesophytic grassland
25	Rokantai, Vilnius distr.	54.73488	25.54795	180	Mesophytic grassland
26	Mickūnai, Vilnius distr.	54.75274	25.55872	156	Pine forest, roadside
27	Užugriovis, Vilnius distr.	54.82782	25.24644	161	Dry grassland
28	Bernatonys, Vilnius distr.	54.90934	25.32271	160	Mesophytic grassland
29	Skardupiai, Šakiai distr.	54.85567	23.03398	47	Mesophytic grassland
30	Dubingiai, Molėtai distr.	55.05911	25.43509	175	Pine forest, roadside
31	Apkartai, Molėtai distr.	55.21658	25.64254	160	Dry grassland
32	Vorėnai, Molėtai distr.	55.35779	25.61012	158	Mesophytic grassland
33	Mikieriai, Anykščiai dist.	55.66058	25.19964	130	Mesophytic grassland
34	Svėdasai, Anykščiai distr.	55.67750	25.36114	111	Dry grassland
35	Maželiai, Rokiškis distr.	56.03197	25.32496	99	Mesophytic grassland
36	Juodžionys, Biržai distr.	56.24353	24.87915	60	Mesophytic grassland

## Data Availability

Data is contained within the article or Appendix A.

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
