# Peer review of "Trends in Phenolic Profiles of *Achillea millefolium* from Different Geographical Gradients"

_plants, 2023, doi:10.3390/plants12040746_

Round 1
Reviewer 1 Report
The manuscript is well prepared and interesting. It concerns the accumulation differences of phenolic acids and flavonoids between Achillea millefolium populations from northern and southern provenance. The experiments are well justified, carefully documented and described sufficiently. The results are well interpreted.
Quantitative data might be provided in the Abstract.
Author Response
We are grateful to the Reviewer for his/her good assessment of our work.
Point 1: Quantitative data might be provided in the Abstract
Response 1: We did not include quantitative data in the Abstract due to limited space. In Abstract was indicated that „Caffeoylquinic acids predominated in leaves, while inflorescences tended to prevail in flavonoids”. “Significant correlation of individual phenolic compounds with latitude-longitude corresponded to their differences between two countries”. In the Abstract is indicated that “the content of phenolic compounds in plant materials from northern latitudes was more than twice that of plants from southern latitudes”.
Reviewer 2 Report
The present study aimed to determine the accumulation differences in phenolic content between geographically distant populations of Achillea millefolium from northern and southern gradients.
Overall, this is well-conceived research with several weaknesses. The Results and Discussion section should be modified. Likewise, the manuscript needs editing in English language and style.
The Authors should decide and hold to either the term “specialized metabolites” or “secondary metabolites” throughout the manuscript.
“Plant parts” should be changed to “plant organs”, since the Authors are analyzing the contest of specific plant organs and not the parts of organs or undefined parts of plants.
The results section should be composed more clearly. It is suggested that the Authors present as a separate first section the total phenolic acid and flavonoid content in different organs from two localities from Table 1 and Table 3, the second section should contain phenolic profiling of A. millefolium flowers, stems, and leaves from Turkey and Lithuania, the third section should have the PCA results.
177 line – Figure 1
Figure 1 – The Authors should emphasize in the figure caption what is marked red and what is green in PCA.
Line 243 – “The main factor associated with phenolic differences at higher altitudes and latitudes is the response to low temperatures, which generally increase UV-B radiation and photooxidative stress.” Could you please make this clearer? How are low temperatures affecting UV-B radiation?
259- controversial results
The Authors have missed discussing the results of phenolic acid contents in different plant organs, which is one of the major focuses of the results of the presented manuscript.
Author Response
The present study aimed to determine the accumulation differences in phenolic content between geographically distant populations of Achillea millefolium from northern and southern gradients.
Overall, this is well-conceived research with several weaknesses. The Results and Discussion section should be modified. Likewise, the manuscript needs editing in English language and style.
The Authors are very grateful to the Reviewer for his/her positive evaluation and valuable comments that we hope helped to improve the manuscript.
Point 1: The Authors should decide and hold to either the term “specialized metabolites” or “secondary metabolites” throughout the manuscript.
Response 1: The term was changed to “specialized metabolites” in all manuscript.
Point 2:“Plant parts” should be changed to “plant organs”, since the Authors are analyzing the contest of specific plant organs and not the parts of organs or undefined parts of plants.
Response 2: The term “plant parts“ was changed to ”plant organs“.
Point 3: The results section should be composed more clearly. It is suggested that the Authors present as a separate first section the total phenolic acid and flavonoid content in different organs from two localities from Table 1 and Table 3, the second section should contain phenolic profiling of A. millefolium flowers, stems, and leaves from Turkey and Lithuania, the third section should have the PCA results.
Response 3: The results section has been recomposed and reorganized based on the Reviewer‘s suggestions.
Point 4: Figure 1 – The Authors should emphasize in the figure caption what is marked red and what is green in PCA.
Response 4: In Fig. 1 scores marked and outlined in brown indicated groups of Turkish populations; scores marked and outlined in green indicated the grouping of Lithuanian populations.
Point 5: Line 243 – “The main factor associated with phenolic differences at higher altitudes and latitudes is the response to low temperatures, which generally increase UV-B radiation and photooxidative stress.” Could you please make this clearer? How are low temperatures affecting UV-B radiation?
Response 5: Revised as follows: The main factor associated with differences in phenolic compounds at high altitudes and high latitudes is the response to low temperatures or increased UV-B radiation, which generally induces photo-oxidative stress [31,32] and promotes the synthesis of phenolic compounds [33].
Point 6: 259- controversial results
Response 6: Revised as follows: Large-scale latitudinal gradient studies have been conducted with a limited number of different species yielding controversy results.
Point 7: The Authors have missed discussing the results of phenolic acid contents in different plant organs, which is one of the major focuses of the results of the presented manuscript.
Response 7: Thank you for the important note. The description of the results of phenolic acid accumulation in different plant organs from two locations was added to section 2.2.
Reviewer 3 Report
The study compares the contents of selected caffeoylquinic acids and flavonoids in leaves, stems, and inflorescences of Achillea millefolium collected in Turkey and Lithuania. It is solid and thorough and presents interesting results.
Several rather minor comments follow:
1. I suggest replacing “distribution patterns” in the title with “profiles” or “contents”.
2. When assessing the differences in the ratio of groups of phenolic compounds between countries, did you pool the data from the different plant parts? If so, why Kruskal-Wallis test was preferred over Mann-Whitney?
3. L290-304 in the discussion section are only loosely connected to the data. I suggest removing the paragraph.
Author Response
The study compares the contents of selected caffeoylquinic acids and flavonoids in leaves, stems, and inflorescences of Achillea millefolium collected in Turkey and Lithuania. It is solid and thorough and presents interesting results.
We are grateful to the Reviewer for the positive assessment of our work and valuable comments.
Several rather minor comments follow:
Point 1: I suggest replacing “distribution patterns” in the title with “profiles” or “contents”.
Response 1: Thank you for the suggestion. The title was modified to: “Trends in phenolic profiles of Achillea millefolium from different geographic gradients”.
Point 2: When assessing the differences in the ratio of groups of phenolic compounds between countries, did you pool the data from the different plant parts? If so, why Kruskal-Wallis test was preferred over Mann-Whitney?
Response 2:
The differences in the total content and ratio of phenolic compound groups between countries were assessed separately for different plant parts (Table 2 and 3).
The major difference between the Kruskal-Wallis and the Mann-Whitney test is that the first can accommodate more than two groups. The parametric equivalent of the Kruskal–Wallis test is ANOVA.Both tests require independent (between-subjects) designs and use summed rank scores to determine the results.
Point 3: L290-304 in the discussion section are only loosely connected to the data. I suggest removing the paragraph.
Response 2: The corresponding paragraph in discussion was deleted.
Reviewer 4 Report
The authors report the finding on analyses of polyphenolic acids and flavonoids, on different part of the plant Achillea millefolium, considering areas at different latitude and longitude. The analyses were carried out in inflorescences, leaves and stems, revealing differences, more or less significant, among the different parts of the plants and the different areas.
I found the study very interesting and well written, although some writing mistakes (report some examples below).
I suggest this manuscript for publication with some minor revisions.
The paragraphs 2.1 and 2.2 can be better organized.
Paragraph: 2.1. Phenolic Profiles of Inflorescences, Leaves and Stems
Some parts of this paragraph may be reported and discussed in the following paragraph, for instance the last part, lines 126-130, compares the different composition between different region, and not different part of the plant. Moreover, it is not clear for lines 128-129: “Small amounts of phenolics were found in the stems, with more notable levels of chlorogenic acid and rutin”, to which plant the authors refer to (Lithuanian or Turkish?). The paragraphs 2.1 and 2.2 can be better organized, mainly moving the last part of 2.1 into 2.2.
Line 52: “both in folk medicine for thousands of years”, “both” has to be removed?
Line 125: “The leaves of all populatioins dominated”, correct “population”, and add The leaves of all populations were dominated”
Line 183: “coffeic”, correct with “caffeic”.
Line 208: “distrant”, correct with “distant”
Line 209: “was contributed by the highest amout of luteolin and apigenin,….”, check the right use of the verb “contribute”, it sounds that this sentence should be better “was due to the highest amount of…”. Correct “amont” with “amount”.
Line 215: “contributed the greatest impact”, correct with “contributed to”
Line 228: “populatoions”, correct with “populations”
Lines 271-273: “Consequently, our study contributes to the understanding of the increased accumulation of phenolic compounds at higher latitudes, but excluding the effect of altitude.”
Why the selected environment may have influenced the production of phenolic compounds, excluding the influence of temperature or high irradiance? The genetic selection is supposed to act maintaining those organisms with phenotypic and physiological characteristics conferring them the capability to live in that specific environment. The phenolic acids and flavonoids are known to be produced in a large amount under stress conditions, in this case low temperature and high irradiance have been shown to be effective. How can the authors explain the fact that at the higher latitude phenols were the highest and at lower ones flavonoids were the highest? This aspect should be better discussed.
Lines 276-279: As previously underlined, the authors say that the highest flavonoids content in the southern population agrees with the UV-b effect at high elevation. However, they previously say that in the areas considered in this study the effect of altitude (different temperature….) can not be considered because there are not significant differences among the altitudes of the considered areas. This point has to be clearest explained.
2.3. Chemical Variation of Populations. Principal Component Analysis
Is it possible for PCA analyse, instead of specific phenolic compounds, also consider the total content of phenols and flavonoids? It should be interesting to analyze the effect of the parameters considered in relation to the total content of phenols and flavonoids, besides the specific compounds. It could be an additive important information. Sometime, some differences can be observed on the quantitative amount of specific compounds, but also on the total amount. The total amount is always higher than the one obtained by HPLC analyses, so the results may input different implications.
Plagiarism has been detected:
Lines 41-47
Author Response
The authors report the finding on analyses of polyphenolic acids and flavonoids, on different part of the plant Achillea millefolium, considering areas at different latitude and longitude. The analyses were carried out in inflorescences, leaves and stems, revealing differences, more or less significant, among the different parts of the plants and the different areas.
I found the study very interesting and well written, although some writing mistakes (report some examples below). I suggest this manuscript for publication with some minor revisions.
We are grateful to the Reviewer for his/her good assessment of our work and valuable comments and suggestions that we hope helped to improve the work.
Point 1: The paragraphs can be better organized.
Paragraph: 2.1. Phenolic Profiles of Inflorescences, Leaves and Stem
Some parts of this paragraph may be reported and discussed in the following paragraph, for instance the last part, lines 126-130, compares the different composition between different region, and not different part of the plant. “Small amounts of phenolics were found in the stems, with more notable levels of chlorogenic acid and rutin”, to which plant the authors refer to (Lithuanian or Turkish?). The paragraphs 2.1 and 2.2 can be better organized, mainly moving the last part of 2.1 into 2.2.
Response 1: subsections 2.1 and 2.2 were highly reorganized and corrected according to the comments.
Point 2: Moreover, it is not clear for lines 128-129:
Response 2: Was corrected to : Small amounts of phenolics were found in the stems of all populations, with more notable levels of chlorogenic acid and rutin in populations from both countries.
Points 3 :
Line 52: “both in folk medicine for thousands of years”, “both” has to be removed?
Line 125: “The leaves of all populatioins dominated”, correct “population”, and add The leaves of all populations were dominated”
Line 183: “coffeic”, correct with “caffeic”.
Line 208: “distrant”, correct with “distant”
Line 209: “was contributed by the highest amout of luteolin and apigenin,….”, check the right use of the verb “contribute”, it sounds that this sentence should be better “was due to the highest amount of…”. Correct “amont” with “amount”.
Line 215: “contributed the greatest impact”, correct with “contributed to”
Line 228: “populatoions”, correct with “populations”
Response 3: Thank you, inaccuracies and mistakes were corrected according to the above comments.
Point 4 : Lines 271-273: “Consequently, our study contributes to the understanding of the increased accumulation of phenolic compounds at higher latitudes, but excluding the effect of altitude.”
Why the selected environment may have influenced the production of phenolic compounds, excluding the influence of temperature or high irradiance? The genetic selection is supposed to act maintaining those organisms with phenotypic and physiological characteristics conferring them the capability to live in that specific environment. The phenolic acids and flavonoids are known to be produced in a large amount under stress conditions, in this case low temperature and high irradiance have been shown to be effective. How can the authors explain the fact that at the higher latitude phenols were the highest and at lower ones flavonoids were the highest? This aspect should be better discussed.
Response 4: The expression is complicated and not exact, because we do not excluding the influence of low temperature at higher latitudes. In our study, elevation was insignificant environmental element at northern latitudes and was not applicable as a comparative factor between the two geographical areas.. Differently, in cited references from Finland higher latitudes were combined with higher altitudes. The correlations of phenolic compounds with latitudes were shown in Table S1
Point 5. Lines 276-279: As previously underlined, the authors say that the highest flavonoids content in the southern population agrees with the UV-b effect at high elevation. However, they previously say that in the areas considered in this study the effect of altitude (different temperature….) can not be considered because there are not significant differences among the altitudes of the considered areas. This point has to be clearest explained.
Response 5: In our study lower latitudes in south were combined with higher elevation and consequently with higher irradiance. There were significant differences for individual phenolics between two regions of Turkey, Nevşehir (1) and Gaziantep (2) provinces (Table S2). No significant correlation was found between the total phenolics content and site elevation (Table S2).
On the other hand, phenolic compound accumulation and the elevation differences between the two Anatolian regions did not show a clearer trend to explain the differences in the accumulation of phenolic compounds between populations.
It can be assumed that the low temperature in northern latitudes has a greater importance for the accumulation of phenolic acids than for flavonoids. On the other hand, the trend of increasing total flavonoid content in southern latitudes can be assumed by increased UV-B radiation at higher elevations compare to northern, suggesting a greater contribution of flavonoids to radical scavenging potential. In this regard, our findings are consistent with previous studies showing the promotion of flavonoid synthesis at high elevation due to increased UV-B radiation and oxidative stress [37].
Point 6: 2.3. Chemical Variation of Populations. Principal Component Analysis.
Is it possible for PCA analyse, instead of specific phenolic compounds, also consider the total content of phenols and flavonoids? It should be interesting to analyze the effect of the parameters considered in relation to the total content of phenols and flavonoids, besides the specific compounds. It could be an additive important information. Sometime, some differences can be observed on the quantitative amount of specific compounds, but also on the total amount. The total amount is always higher than the one obtained by HPLC analyses, so the results may input different implications.
Response 6: PCA2 model showed the same trend for the distribution of Turkish and Lithuanian populations, which falls into two separate groups as in the PCA1. On the other hand, the scores clustered less tightly on PCs plot displayed greater variation than in the PCA1 model using selected individual phenolic compounds. Point 7: Plagiarism has been detected:Lines 41-47.
Response 7: the paragraph was rewritten.
Reviewer 5 Report
Dear Editors,
The manuscript Distribution patterns of Achillea millefolium phenolic compounds in different geographical gradients with authors Jolita Radušienė, Birutė Karpavičienė, Lina Raudone, Gabriele Vilkickyte, Cüneyt Cirak, Fatih Seyis, Fatih Yayla Mindaugas Marksa, Laura Rimikeė, Liudas Ivanauskas is, in my opinion, very interesting and original. The plant that the authors studied is known to many researchers. What is interesting in this manuscript is that research was done in two different geographical regions under different climatic and temperature conditions. The authors draw the readers' attention to the fact that the environment has an influence on phenolic compounds. The authors, with the results obtained by them, emphasize that northern latitudes have a beneficial effect on the medicinal plant Achillea millefolium
Let the authors pay attention to the introduction part. I think the sources used are old and there are only 6 from the last 3-5 years. I request the authors to add more new sources.
As for the results part, their description is well structured and gives a clear idea. Please let the authors, when entering abbreviations for the first time, write the full name, for example DM and others.
The discussion thoroughly analyzes the obtained results and allows the authors to present clear and precise conclusions to the readers.
In my opinion, in the methods and materials part, they have presented everything that is needed. I think that the methods of analysis that the authors have used are well presented and could be replicated by any researcher.
The conclusions correctly reflect the results obtained and the conclusions reached from the discussions.
Author Response
The manuscript Distribution patterns of Achillea millefolium phenolic compounds in different geographical gradients with authors Jolita Radušienė, Birutė Karpavičienė, Lina Raudone, Gabriele Vilkickyte, Cüneyt Cirak, Fatih Seyis, Fatih Yayla Mindaugas Marksa, Laura Rimikeė, Liudas Ivanauskas is, in my opinion, very interesting and original. The plant that the authors studied is known to many researchers. What is interesting in this manuscript is that research was done in two different geographical regions under different climatic and temperature conditions. The authors draw the readers' attention to the fact that the environment has an influence on phenolic compounds. The authors, with the results obtained by them, emphasize that northern latitudes have a beneficial effect on the medicinal plant Achillea millefolium
As for the results part, their description is well structured and gives a clear idea. The discussion thoroughly analyzes the obtained results and allows the authors to present clear and precise conclusions to the readers.
In my opinion, in the methods and materials part, they have presented everything that is needed. I think that the methods of analysis that the authors have used are well presented and could be replicated by any researcher. The conclusions correctly reflect the results obtained and the conclusions reached from the discussions.
Authors are very grateful to the Reviewer for his/her positive assessment of presented study.
Point 1: Let the authors pay attention to the introduction part. I think the sources used are old and there are only 6 from the last 3-5 years. I request the authors to add more new sources.
Response: some new sources were added to references.
Point 2: Please let the authors, when entering abbreviations for the first time, write the full name, for example DM and others.
Response: the abbreviations were written in full for the first time mention.
Reviewer 6 Report
The authors aim to determine the differences in accumulation of phenolic compounds between geographically distant populations of Achillea millefolium from northern and southern gradients (from Gaziantep and Nevşehir provinces in Turkey and from wild populations in Lithuania). A complex of nine hydroxycinnamic acids and eleven flavonoids was identified and quantified in the methanolic extracts of inflorescences, leaves and stems using the HPLC-PDA method. The PCA score plot model was used to represent the quantitative distribution pattern of phenolic compounds along geographical gradient of populations. The authors consider the work important because there are no available data on differences in the accumulation of phenolic compounds in plant parts of A. millefolium depending on the geographical gradient of the population site and their study has potential for planning the targeted use of plant materials and managing the overexploitation of plant resources.
The manuscript is well structured and the methods and data are clearly presented.
Observations:
- How polyphenolic acids and flavonoids from table 1 were determined?
- A brief discussion of the therapeutic importance of the predominant individual compounds would be necessary (in the discussions, after the penultimate paragraph).
Please check the spelling and grammar in all the manuscript and the dimension of font.
Author Response
The authors aim to determine the differences in accumulation of phenolic compounds between geographically distant populations of Achillea millefolium from northern and southern gradients (from Gaziantep and Nevşehir provinces in Turkey and from wild populations in Lithuania). A complex of nine hydroxycinnamic acids and eleven flavonoids was identified and quantified in the methanolic extracts of inflorescences, leaves and stems using the HPLC-PDA method. The PCA score plot model was used to represent the quantitative distribution pattern of phenolic compounds along geographical gradient of populations. The authors consider the work important because there are no available data on differences in the accumulation of phenolic compounds in plant parts of A. millefolium depending on the geographical gradient of the population site and their study has potential for planning the targeted use of plant materials and managing the overexploitation of plant resources.
The manuscript is well structured and the methods and data are clearly presented.
We are grateful to the Reviewer for his good assessment of our work and comments.
Point 1:- How polyphenolic acids and flavonoids from table 1 were determined?
Response 1: A nine phenolic acids and eleven flavonoids was identified and quantified in A. millefolium inflorescences leaves and stems. The quantities of all compounds of each of these groups were summed.
Point 2- A brief discussion of the therapeutic importance of the predominant individual compounds would be necessary (in the discussions, after the penultimate paragraph).
Response 2: The brief discussion on phenolic compounds was added in the discussion.
Point 3: Please check the spelling and grammar in all the manuscript and the dimension of font.
Response: the spelling and the font dimension was checked in all the manuscript.